# Minimally Invasive Surgery for Posterior Spinal Instrumentation and Fusion in Adolescent Idiopathic Scoliosis: Current Status and Future Application

**DOI:** 10.3390/children10121882

**Published:** 2023-11-30

**Authors:** Ludmilla Bazin, Alexandre Ansorge, Tanguy Vendeuvre, Blaise Cochard, Anne Tabard-Fougère, Oscar Vazquez, Giacomo De Marco, Vishal Sarwahi, Romain Dayer

**Affiliations:** 1Division of Paediatric Orthopaedics, Faculty of Medicine, Geneva University Hospitals, 1211 Geneva, Switzerlandoscar.vazquez@hcuge.ch (O.V.);; 2Department of Spine Surgery, Lucerne Cantonal Hospital, 6000 Lucerne, Switzerland; 3Department of Orthopedic and Trauma Surgery, University Hospital of Poitiers, 86000 Poitiers, France; 4Department of Pediatric Orthopedics, Cohen Children’s Medical Center, Northwell Health, New Hyde Park, New York, NY 11040, USA

**Keywords:** adolescent idiopathic scoliosis, correction, posterior instrumentation and fusion, paraspinal muscle approach

## Abstract

The posterior minimally invasive spine surgery (MISS) approach—or the paraspinal muscle approach—for posterior spinal fusion and segmental instrumentation in adolescent idiopathic scoliosis (AIS) was first reported in 2011. It is less invasive than the traditionally used open posterior midline approach, which is associated with significant morbidity, including denervation of the paraspinal muscles, significant blood loss, and a large midline skin incision. The literature suggests that the MISS approach, though technically challenging and with a longer operative time, provides similar levels of deformity correction, lower intraoperative blood loss, shorter hospital stays, better pain outcomes, and a faster return to sports than the open posterior midline approach. Correction maintenance and fusion rates also seem to be equivalent for both approaches. This narrative review presents the results of relevant publications reporting on spinal segmental instrumentation using pedicle screws and posterior spinal fusion as part of an MISS approach. It then compares them with the results of the traditional open posterior midline approach for treating AIS. It specifically examines perioperative morbidity and radiological and clinical outcomes with a minimal follow-up length of 2 years (range 2–9 years).

## 1. Introduction

Adolescent idiopathic scoliosis (AIS) is the most common spine deformity in the adolescent population. Its prevalence in most populations is about 2.5% [1,2,3,4]. Approximately 0.1 to 0.25% of AIS patients eventually undergo surgical treatment when they exceed a certain Cobb angle threshold [1,5,6].

Posterior spinal fusion (PSF) and segmental spinal instrumentation (SSI) using pedicle screws is the most frequently used surgical technique for treating AIS [7,8]. It was first reported by Suk et al. in 1995 and further supported by their later publication (2001) of the first large retrospective series of pediatric deformity cases operated on using this technique and an open posterior midline approach [9,10]. At that time, it was rarely used because of fears of causing neurological damage secondary to poorly positioned pedicle screws. Suk et al.’s series included 462 patients with a deformity (330 idiopathic scoliosis cases) who were operated on using 4604 pedicle screws [9]. As no significant neurological or visceral complications adversely affecting the long-term outcomes were observed, they considered the technique to be reliable and safe. It was associated with significant deformity correction (72%) and reliable correction maintenance (1% correction loss). Posterior segmental pedicle screw instrumentation gained popularity in the 2000s, as evidence was showing the superiority of deformity correction and maintenance of correction, leading to reduced revision surgeries and reduced need to perform additional anterior release surgeries for correcting large curves when compared to previous fixation techniques, like hook-based instrumentations [5,8,11,12].

Ten years later, Sarwahi et al. published a surgical technique paper including two case reports of AIS patients operated on using a posterior paraspinal muscle approach through three small skin incisions—the minimally invasive spine surgery (MISS) approach—to perform PSF and SSI using pedicle screws [13]. As the two initial cases of AIS reported by Sarwahi et al. seemed to reach coronal and sagittal deformity corrections comparable to PSF and standard open SSI, MISS appeared to be a feasible surgical option. They hypothesized multiple potential advantages associated with using this new posterior MISS approach compared to the routine open posterior midline approach, including less blood loss, shorter hospital length of stay, less pain, and the concurrent need for less pain medication, based on the emerging evidence supporting minimal invasive spine surgery for treating adult spine deformities [14,15].

Since MISS for AIS was first introduced in 2011, multiple case series and comparative series, as well as two meta-analyses, evaluated the degree of deformity correction and the potential advantages of this technique in comparison to the traditional open posterior midline approach [13,16,17,18,19,20,21,22,23,24,25,26,27,28,29,30].

However, not all the relevant available evidence has been comprehensively summarized in a review until now. Therefore, this narrative review describes the posterior MISS approach for performing PSF and SSI on AIS patients and compares its perioperative morbidity and radiological and clinical outcomes with those obtained using the traditional open posterior midline approach. The majority of the cited studies do not select specific Lenke types of curves. Should this be the case, it is explicitly stated where appropriate.

## 2. Surgical Technique

Wiltse et al. first described the paraspinal muscle approach in 1968 [31]. In 1988, they reported changes to their approach in order to use it for treating additional conditions such as lumbar disc herniations, spinal stenosis and spondylolisthesis in adult patients [32]. The original Wiltse approach involved two paramedian skin incisions with bilateral paramedian incisions of the thoracolumbar fascia and bilateral blunt dissections to separate the multifidus and longissimus muscles. This approach allows for direct access to the lumbar spine’s articular processes, laminas, pars interarticularis, and transverse processes.

To minimize skin disruption for cosmetic reasons, this soft-tissue-sparing approach was modified by Sarwahi et al. for use in AIS patients [24]. Instead of using two long paramedian skin incisions, three shorter midline skin incisions are made. The locations of these incisions are determined by the deformity and the resulting preoperative plan for pedicle screw positioning. Fluoroscopy is used preoperatively to mark the incision locations on the skin surface (Figure 1a). Usually, two to five vertebrae are instrumented through each skin incision, and one to two vertebrae are left with no instrumentation between them. Subcutaneous fat in the thoracic region is sharply dissected along the midline, the trapezius muscle, and the latissimus dorsi muscle. The rhomboid minor and major muscles, together with their fascial attachments, are separated from the spinous processes and retracted laterally to allow for a paramedian incision in the thoracolumbar fascia. The extent to which these superficial muscles need to be sharply dissected depends on the exact location of the three incisions and the number of levels to be instrumented and fused. Subcutaneous fat in the lumbar region is directly undermined laterally to allow for a paramedian incision in the thoracolumbar fascia. This is followed by a blunt muscle-sparing approach used to reach the lumbar spine’s facet joints—the transverse processes in the thoracic spine (Figure 1b). Gelpi retractors are usually used for this approach, but some surgeons use tubular retractors; both techniques permit delicate muscle dissection and are believed to be equivalent [33]. Ultimately, the posterior elements are exposed from the base of the laminas to the transverse processes using electrocautery. The exposure described here can only be performed on one side at a time. It is followed by the performance of ipsilateral wide facetectomies, with cartilage removal using a bone chisel or a high-speed burr, cannulation of the ipsilateral pedicles using the freehand technique, and the insertion of pedicle markers into the pedicle channels. These steps are then repeated on the other side. If computerized tomography (CT)-based navigation is used instead of the routinely used freehand technique, the posterior bony elements do not need to be exposed using electrocautery [25]. A mixture of autografts from the facetectomies and freeze-dried allograft bone is then applied over the decorticated facet joints. The facet joints between the skin incisions are also decorticated and fused on both sides. Next, the pedicle markers are replaced by pedicle screws using guide wires on one side (the convex side of the major curve is usually addressed first), and then a cobalt–chrome rod, contoured to reproduce the appropriate thoracic kyphosis and lumbar lordosis, is inserted into the reduction tubes fixed on the pedicle screw heads (Figure 2). Depending on the surgeon’s preference, the rod can be inserted caudally to reduce the risk of intrusion into the spinal canal or cephalad to avoid inadvertently pushing on the patient’s head. Gradual spine-to-rod reduction, using reduction tubes, is used to correct most of the deformity. When additional deformity correction is needed, an additional direct apical segmental derotation is then performed. After the rod’s definitive fixation to the screw heads, the reduction tubes are removed. The opposite side is than similarly instrumented. If the amount of correction still needs to be increased at this point, adequately contouring the second rod might enable additional deformity correction through spine-to-rod reduction. Finally, the paraspinal muscle approach is sutured using a routine layered technique. In 2019, Urbanski et al. reported a modification of the paraspinal muscle approach which further reduced soft tissue disruption [25]. Their technique used percutaneous, trans-muscular stab incisions to access the pedicle entry points. As no posterior bony landmarks are exposed, this technique requires CT-based navigation. The latter technique is known to achieve higher pedicle screw placement accuracy and exposes the patients to roughly four times more radiation than the freehand technique (effective dose between 1.11 and 1.48 mSv versus 0.17 and 0.34 mSV), while the rates of pedicle screw misplacement-related complications (0–1.4%) are similar for both techniques [34,35,36,37,38,39].

Another modification of the paraspinal muscle approach was reported by Sarwahi et al. in 2023. Its only change involved replacing the three skin incisions with a single, longer skin incision. Compared to the three-incision paraspinal muscle approach (the original MISS), the operative time was shorter and the advantages over an open posterior midline approach were maintained [30].

## 3. Deformity Correction and Fusion

### 3.1. Coronal Correction

Most reports support the view that performing SSI and PSF with pedicle screws using a posterior MISS approach results in coronal deformity corrections that do not differ significantly from those obtained using a standard posterior midline approach (see Table 1) [23,24,25,26,27,28,30]. The best available evidence for this view is the 2022 meta-analysis by Yang et al. [30] They analyzed five comparative series for this parameter, including 713 patients, and found a weighted mean difference (WMD) of −0.01 (95% CI −0.03 to 0.01; *p* = 0.518) [30]. The follow-up (FU) lengths of these series varied between 2 and 9 years.

However, three moderately-sized comparative retrospective series have been inconsistent with this view. The first was published by Miyanji et al. in 2015 and included 46 AIS cases with an FU length of 2 years [23]. It reported a coronal curve correction rate of 58% in the MISS group and 68% in the open posterior midline approach group (*p* < 0.001). The authors thought that this correction difference might be explained by the new technique’s learning curve effect. The second series, including 49 patients, was published by Yang et al. in 2021. It found a statistically significant approach-related difference in the coronal major curve correction of 5% (*p* = 0.017) and an approach-related postoperative major curve Cobb angle difference of 3° [26]. A correction curve difference of 3° might not be clinically significant or related to the selected approach, but rather to the correction technique used. Indeed, monoaxial screws with direct apical vertebral derotation were used in conjunction with the open posterior midline approach, but polyaxial screws with spine-to-rod translation were the only reduction technique used in conjunction with the MISS approach. The third series, including 82 patients with Lenke type 1 curves, was described by Syundyukov et al. in 2023 [40]. The coronal major curve correction was significantly greater in the open posterior midline group than in the MISS group when expressed in percentage terms (88% vs. 78%; *p* < 0.001), but not when expressed in degrees (40.5° vs. 46.7°; *p* = 0.005).

### 3.2. Sagittal Correction

When treating AIS cases, spine surgeons have traditionally focused mainly on correcting coronal deformities [42]. Over the last 20 years, evidence has grown concerning the importance of the physiological sagittal balance, which is necessary to maintain a pain-free erect posture. Consequently, more attention is now given to restoring the patient’s physiological sagittal profile, and particularly to correcting the typically encountered thoracic hypokyphosis present with major thoracic curves [42,43,44,45]. The two first comparative series, published by Miyanji et al. and Sarwahi et al., reported no significant differences in sagittal deformity correction between their MISS and open posterior midline approach groups [23,24]. Interestingly, the five studies included in Yang et al.’s meta-analysis, which evaluated the sagittal correction, revealed a significant difference in the correction rate for thoracic kyphosis [30]. At their last follow-up, which varied between 2 and 9 years, the pooled MISS and the pooled open posterior midline groups had mean thoracic kyphosis values of 25.80° and 22.71°, respectively. This difference appeared to be especially significant among patients with more than 10 levels fused. In a comparative study including 485 AIS cases with a minimal FU length of 2 years (range of 2–5 years), Sarwahi et al. again found a significantly greater kyphosis correction among MISS patients than among open posterior midline approach patients (kyphosis increase of 17.9% versus −5.3%; *p* = 0.007) [28]. This finding is difficult to explain. It might be related to better preservation of the paraspinal muscles and the posterior ligament complex, which resist the lordosing effect of the direct apical vertebral derotation technique that is often used to correct scoliosis. Indeed, the major forces applied during this maneuver push the thoracic hump ventrally to decrease the rotational deformity and concomitantly induce a reduction in the thoracic kyphosis, as previously reported by Sudo et al. [45]. This explanation, however, contradicts the general understanding that an extensive posterior release enables better restoration of kyphosis [28].

Only the 2023 case series described by Syundykov et al. found a significantly better correction of thoracic hypokyphosis using the open posterior midline approach than using the MISS approach [40]. Their series of 82 patients with Lenke type 1 curves showed a mean increase in the thoracic kyphosis of 4° using the open posterior midline approach and a mean decrease of 4° using the MISS approach. They related the better thoracic kyphosis correction obtained using the midline approach to its better access to the facet joints and to the more extensive ligamentous release.

In summary, there is more evidence supporting the MISS approach as the best one for thoracic kyphosis restoration.

### 3.3. Fusion Rate

In clinical practice, fusion assessment is usually performed following an analysis of anteroposterior and lateral standing whole-spine radiographs, as routinely performing CT scans would expose adolescents to unnecessarily high doses of radiation. In cases of significant postoperative back pain, with or without radiological signs of pseudarthrosis, a CT scan is usually performed for a more detailed assessment of fusion status and possible implant-related complications. In 1994, Bridwell et al. described a fusion status classification based on standing X-rays (anteroposterior and lateral views) that is still commonly used in research articles [46]. They rated the fusion mass as “definitely solid” (with heavy trabeculations seen along the whole length of the fusion), “probably solid” (meaning there was no evidence of instrumentation failure or a loss of correction, but that mature trabeculation could not be identified at every level), or as “definite pseudarthrosis” (defined as instrumentation failure or a loss of correction greater than 10°, or visible pseudarthrosis).

In a recent comparative study by Yang et al. [33], fusion rates were assessed after a mean FU length of 22 months (range 18–38 months) using Bridwell’s classification rating on 86 AIS patients operated on using either an open posterior midline approach with SSI and posterior fusion with allografts or an MISS approach. The MISS group was divided into three subgroups based on the bone substitute used: allograft versus demineralized bone matrix versus demineralized cancellous bone chips. CT scans were only performed on patients with back pain or neurological abnormalities, and were also reviewed to determine fusion status. A “definitely solid” or “probably solid” fusion was achieved in 83% of the MISS group patients and 97% of the posterior midline approach group patients (*p* = 0.07). The bone substitute type which was used did not significantly influence the fusion rate in the three MISS subgroups (85% for allograft, 100% for demineralized bone matrix, and 100% for demineralized cancellous bone chips; *p* = 0.221).

In their meta-analysis, Yang et al. noted the diversity of patients in terms of their curve types and fusion levels across the various studies [30]. Some studies focused on specific Lenke types, while others included a mix of curve types (Lenke types 1–6), and the fusion levels ranged widely from 5 to 12. This heterogeneity complicates direct comparisons of fusion success according to the approach used. Despite these complexities, the occurrence of hardware failures, such as screw or rod breakage, was not significantly different between the MISS and open posterior midline approach groups. This suggests that both approaches can achieve comparably high levels of hardware stability and fusion rates.

## 4. Perioperative Morbidity

### 4.1. Estimated Blood Loss and Allogeneic Transfusion Rate

Correcting AIS using SSI and PSF by means of an open posterior midline approach is associated with extensive subperiosteal preparation and a large wound surface. In contrast, the posterior paraspinal muscle approach—the MISS approach—is associated with much less soft tissue disruption. It might, therefore, significantly decrease the mean estimated blood loss (EBL) and the need for allogeneic blood transfusions. Multiple comparative studies have indeed shown significantly lower EBL when using MISS than when using an open posterior midline approach (see Table 2). For instance, in their series of eight AIS cases with Lenke type 5C curves, Urbanski et al. reported a mean EBL of 138 mL when using MISS versus 450 mL (*p* = 0.016) when using the open posterior midline approach [25]. Their particular low EBL might have been associated with their use of CT-navigation in conjunction with fascial stab incisions, allowing for further minimization of soft tissue disruption, as no bony landmarks needed to be exposed. Yang et al.’s comparative series, including 49 AIS patients, reported a much higher mean EBL with both techniques, but their MISS group still had a significantly lower mean EBL than their open posterior midline group (1279 mL versus 2503 mL, respectively; *p* < 0.001) [26]. In 2023, Sarwahi et al. reported a large comparative series of 532 AIS cases operated on using either an open posterior midline approach (294 cases), the original three-incision MISS approach (179 cases), or a modified MISS approach known as single long-incision minimally (SLIM) invasive surgery (59 cases) [41]. The mean EBL for the open posterior midline approach group (500 mL) was significantly higher than for the two other groups (302 mL versus 325 mL, respectively; *p* < 0.00001). The allogeneic transfusion rate (19% versus 5.6% versus 6.8%, respectively; *p* = 0.001) was also significantly higher for the open posterior midline approach group than for the two other groups. Interestingly, the original MISS group and the SLIM group had comparable mean EBL values (302 mL versus 325 mL, respectively) and allogeneic transfusion rates (5.6% versus 6.8%), suggesting that the extent of the approach-related muscle dissection is more closely associated with the amount of blood loss than the skin incision length. The strongest current evidence corroborating the lower mean EBL when using MISS can be found in Yang et al.’s 2022 meta-analysis, which included six studies and a total of 767 patients [30]. They reported a mean EBL of 288 mL for the MISS group versus 517 mL for the open posterior midline approach group. The same meta-analysis also reported a significantly lower allogeneic blood transfusion rate in the MISS group than in the open posterior midline approach group (8.0% versus 35.0%, respectively; *p* < 0.001) when analyzing the pooled results of the four studies they included to provide data on allogeneic transfusions.

### 4.2. Operative Time

The MISS approach exposes significantly fewer posterior spinal bony landmarks than the open posterior midline approach. This makes MISS more challenging than the open posterior midline approach when using the freehand technique to perform SSI with pedicle screws. Also, because the skin incisions are on the midline and need to be retracted laterally to one side to perform SSI, instrumentation cannot be carried out bilaterally at the same time, as opposed to with the open posterior midline approach. As a consequence, MISS usually requires a significantly longer mean operative time (ORT) (7.4 to 8.98 h) than the open posterior midline approach (5.77 to 7.07 h) [25,26,27,28,41]. This was especially true in the first reported series, which was also influenced by the learning curve effect [47]. Indeed, the early series reported by Sarwahi et al. showed much longer ORTs for MISS approaches than for open posterior midline approaches (8.98 versus 7.07 h, respectively; *p* = 0.011), as did Miyanii et al. (475.3 versus 346.4 min, respectively; *p* = 0.000) [13,22]. The meta-analysis by Yang et al. showed consistently longer ORTs (89 min longer) for MISS approaches than for the open posterior midline approach [30]. To address this disadvantage of the original MISS approach, Sarwahi et al. recently developed and reported a modification to it consisting exclusively of the replacement of the three short skin incisions with a single longer skin incision (SLIM). In their comparative series, ORT was reduced to 262 min when using SLIM compared to 302 using the original MISS approach with three short incisions, while the open posterior midline approach’s ORT was 258 min [41].

### 4.3. Postoperative Pain and Average Opioid Consumption

The Scoliosis Research Society 22-item (SRS-22) pain score and the Visual Analogue Scale (VAS) score are the most direct ways to report pain. The degree of postoperative opioid consumption can also be used to report pain indirectly. The first series reporting MISS use for treating AIS did not find a significant decrease in pain when using MISS in comparison to the use of the open posterior midline approach (average VAS score 3.5 versus 3.4, respectively; *p* = 0.698) [24]. In contrast, the majority of later series reported lower VAS scores or better SRS-22 pain scores for MISS than for the open posterior midline approach [26,27,41]. This fact is further supported by the results of the meta-analysis by Yang et al. [30]. Indeed, the pooled results of the five studies reporting it revealed significantly less postoperative pain according to the VAS score (WMD, 0.84; 95% CI 0.03 to 1.64; *p* = 0.042) and the SRS-22 pain score (WMD, 0.53; 95% CI 0.06 to 1.00; *p* = 0.02). The large comparative series reported by Sarwahi et al., including 485 AIS patients, analogously reported lower postoperative opioid consumption in their MISS group than in their open posterior midline approach group (*p* < 0.001) [28].

### 4.4. Hospital Length of Stay (LOS)

Hospital length of stay (LOS) is an important indirect marker of postoperative pain and function and has significant financial implications. To the best of our knowledge, only the first series of MISS use reported by Sarwahi et al., which included 22 AIS cases, failed to show significantly a shorter LOS for MISS than for the open posterior midline approach (*p* = 0.472), which might be related to the small number of patients or to the learning curve effect [24]. In contrast, later studies have consistently demonstrated otherwise. For example, the comparative series reported by Urbansky et al., which included only Lenke type 5C curves, showed a significantly shorter LOS for MISS (3.75 versus 7 days; *p* = 0.043) [25]. The results of the meta-analysis by Yang et al. further support this finding (WMD, -1.48; 95% CI −2.48 to −0.48; *p* = 0.004) [30]. Sarwahi et al. found no significant difference in LOS between their original MISS technique with three small skin incisions and their more recent modification with one long skin incision (4 days for both techniques; *p* = 0.7). The LOS was still significantly longer for their open posterior midline group (5 days; *p* < 0.001) [41].

### 4.5. Intraoperative, Perioperative, and Long-Term Complications

Various complications related to the surgical correction of AIS have been defined and reported in the literature. Hariharan et al. reported the largest 10-year prospective follow-up study to evaluate postoperative complications after the surgical treatment of AIS patients [48]. Of the 282 patients included, 195 underwent posterior spinal fusion using an open posterior midline approach. A total of 19 complications occurred in 18 of the 195 patients (9.7% complication rate), with the most prevalent being surgical site infections (37%), followed by adding-on (26%), pulmonary (16%), neurological (11%), instrumentation (5%), and gastrointestinal issues (5%).

When comparing the complication rates after SSI and PSF using either MISS or the open posterior midline approach, the available comparative series found no statistically relevant differences [23,24,26,27,28,29,41]. For instance, the largest case–control series, comparing 192 MISS cases to 293 open posterior midline approach cases, showed similar perioperative complication rates (≤30 days) among both groups (3.1% versus 3.8%; *p* = 0.81) [28]. This was also the case with long-term complications (>30 days) (3.6% versus 1.4%; *p* = 0.12) after a minimal FU length of 2 years (range 2–5 years). Likewise, Yang et al. found no significant approach-related complication rate differences in their meta-analysis (RR, 1.13; 95% CI 0.77 to 1.67; *p* = 0.521), which defined surgical site infection, hardware failure, wound dehiscence, pseudarthrosis, and hemothorax as possible complications [30]. Thus, MISS seems to be a safe alternative to the open posterior midline approach.

## 5. Clinical and Functional Outcomes

The available literature has usually measured clinical and functional outcomes using the SRS-22 questionnaire. At the two-year follow-up point, Miyanji et al. observed no differences in SRS-22 outcome scores between AIS patients operated on using either the open posterior midline or MISS approaches (*p* = 0.715) [23]. Yang et al. found similar findings in their comparative series at a mean FU length of 9.7 versus 4.6 years for their MISS and open posterior midline approach groups, respectively [26]. Their meta-analysis found non-statistically-significant but slightly higher SRS-22 scores for self-image/appearance and overall satisfaction among patients who underwent MISS [30]. In their comparative series, including 112 AIS cases (Lenke type 1–4 curves) with a minimum follow-up of two years, Si et al. observed lower SRS-22 pain scores in the MIS group than in the PSF group (*p* = 0.043), and found no significant differences in the other SRS-22 score components at the last follow-up (31 versus 32 months FU for the MISS and the open posterior midline group, respectively) [27]. Sarwahi et al. matched 50 AIS patients operated on using the original MISS approach, with 50 patients operated on using the modified single-incision MISS approach and 50 patients operated on using the open posterior midline approach [41]. They were matched according to age, sex, body mass index, and number of levels fused. At 5–6 months of follow-up, the three groups’ overall SRS-22 questionnaire scores showed no statistical differences. In contrast, the SRS-22 function and activity scores and pain scores were significantly better for the two MISS groups than for the open posterior midline approach group. On the Sports Activity Questionnaire, MISS patients (both groups) were more likely to return to non-contact (*p* = 0.0096) and contact sports (*p* = 0.0095) within 6 months than the patients operated on using the open posterior midline approach. Considering the relevant available reports, MISS seems—at the very least—not to be inferior to the traditional open posterior midline approach in terms of clinical and functional outcomes at 6 months or 2 years of follow-up. The more recent reports, which have analyzed larger patient cohorts, tend to show the MISS approach’s superiority over the posterior open midline approach.

## 6. Conclusions

Segmental spinal instrumentation (SSI) with pedicle screws and posterior spinal fusion (PSF) using an open posterior midline approach is the most commonly used surgical technique to treat adolescent idiopathic scoliosis (AIS). Based on several comparative series including up to 532 patients and a meta-analysis including 767 patients, the newer minimally invasive spinal surgery (MISS) approach, first introduced by Sarwahi et al. in 2011, appears to be an appropriate alternative to the open posterior midline approach for performing SSI and PSF to treat AIS of any Lenke type. The MISS approach has notably been shown to result in equivalent coronal deformity correction, with some evidence supporting better restoration of thoracic kyphosis. MISS also achieves equivalent complication rates and fusion rates. The relevant advantages of MISS over the open posterior midline approach are lower estimated blood loss, lower perioperative allogeneic transfusion rates, less postoperative pain, and a shorter length of stay at the hospital. The clinical and functional outcomes reported for MISS patients at FU lengths varying between 2 and 9 years are at least as good as those obtained using the open posterior midline approach, while some evidence supports a faster return to non-contact and contact sports among MISS patients.

However, the posterior MISS approach also has limitations. As the exposure is restricted in comparison to the traditional posterior midline approach, it is technically more challenging and associated with longer ORT. We, therefore, recommend that surgeons willing to adopt it exclude cases with major curves over 70° or with less than 50% flexibility during the learning curve period. According to Yang et al., which evaluated this learning curve effect in a recent case series including 76 AIS patients, a trained surgeon for conventional open scoliosis surgery needs to operate 46 times using the MISS technique to achieve proficient surgical skills.

Finally, MISS is a safe, effective alternative to the open posterior midline approach and appears to be superior in terms of perioperative morbidity. We, therefore, encourage surgeons to re-evaluate their routine approaches to SSI and PSF in favor of the MISS approach. In this context, using the single-long-incision, minimally (SLIM) invasive surgery technique provides a valid and more easily generalizable alternative. It significantly shortens the total operative time and reduces the technical complexities associated with the original MISS procedure while preserving the other advantages for AIS patients.

## Figures and Tables

**Figure 1 children-10-01882-f001:**
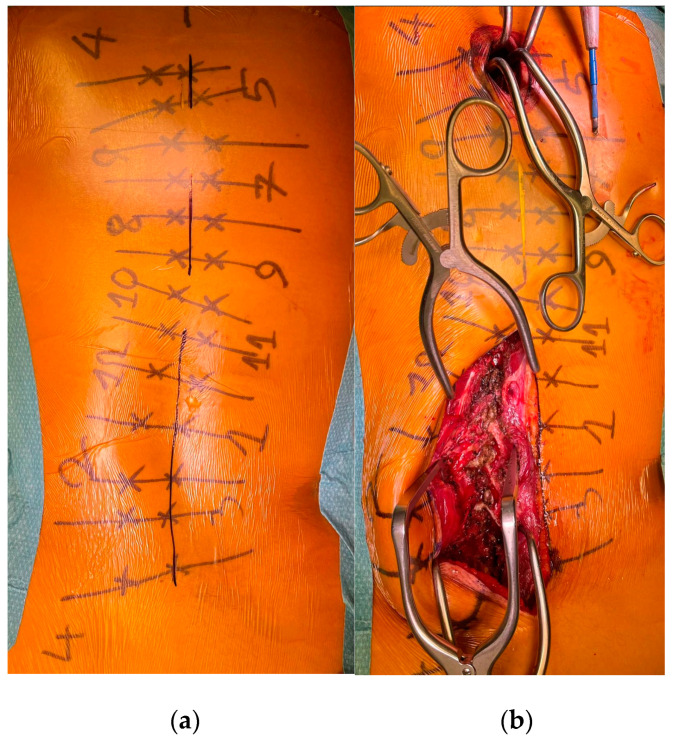
(**a**) Preoperative skin marking of the vertebrae, the pedicles, and the three skin incisions. (**b**) MISS exposure performed on the left lumbar area, with exposure of the facet joints.

**Figure 2 children-10-01882-f002:**
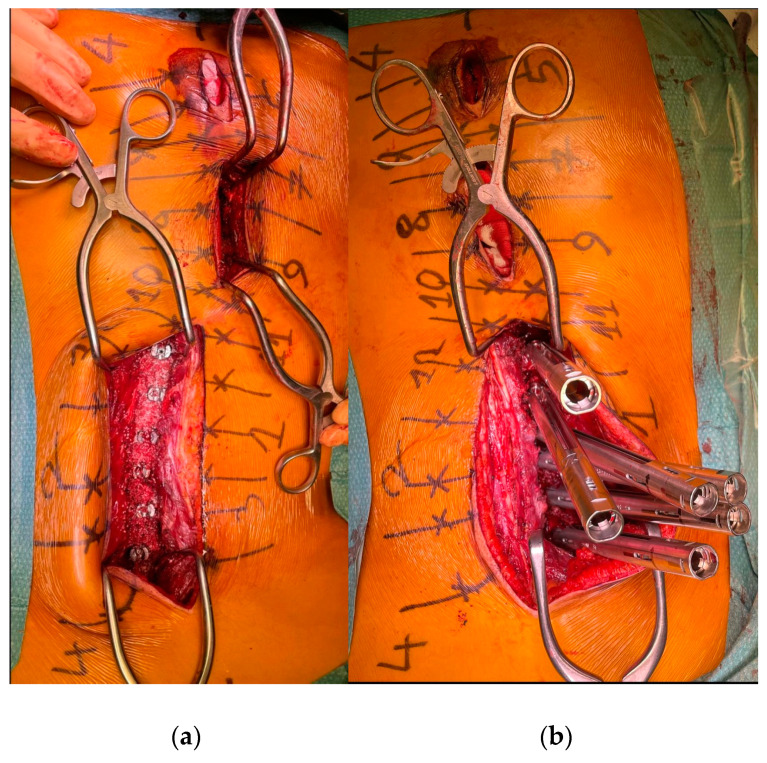
(**a**) The pedicle screws are in place. (**b**) The reduction tubes are fixed on the pedicle screw heads.

**Table 1 children-10-01882-t001:** Coronal major curve correction rates (%) among AIS patients who underwent posterior SSI and PSF using MISS or the open posterior midline approach.

Authors	Year	Study Design	No. of Cases	MISS (%)	OM (%)	*p*-Value
Miyanji et al. [22]	2013	Pros. comp.	32	63	68	n/a ^a^
Miyanji et al. [23]	2015	Retro. comp.	46	58	68	0.001
Sarwahi et al. [24]	2016	Retro. comp.	22	79	85	0.503
Urbanski et al. [25]	2019	Retro. comp.	8	68 ^b^	78	0.072
Yang et al. [26]	2021	Retro. comp.	49	65	70	0.017
Si et al. [27]	2021	Retro. comp.	112	65	64	0.862
Sarwahi et al. [28]	2021	Retro. comp.	485	69	68	0.46
Syundyukov et al. [40]	2023	Retro. comp.	82	78	88	<0.001
Sarwahi et al. [41]	2023	Retro. comp.	532	69 or 62 ^c^	68	0.49
Yang et al. [30]	2022	Meta-analysis	713	n/a ^d^		0.518

The references number [25,27,40], specifically selected Lenke curves type 5C, 1-4, respectively 1. AIS = adolescent idiopathic scoliosis; SSI = segmental spinal instrumentation; PSF = posterior spinal fusion; MISS = minimally invasive spinal surgery; No. = number; OM = open posterior midline approach; pros. comp. = prospective comparative series; retro. comp. = retrospective comparative series; n/a = not available; CI = confidence interval; ^a^ = not statistically significant (95% CI −0.12 to 0.04); ^b^ = modified MISS technique using a single midline skin incision instead of three and fascial stab incisions for performing CT-navigated SSI; ^c^ = 69% correction rate using the original MISS technique with three midline skin incisions or 62% correction rate using the modified MISS technique with a single midline skin incision. ^d^ = correction rates expressed as a WMD of −0.01; 95% CI −0.03 to 0.001.

**Table 2 children-10-01882-t002:** Mean EBL (ml) among AIS patients who underwent posterior SSI and PSF using MISS or the open posterior midline approach.

Authors	Year	Study Design	No. of Cases	MISS (mL)	OM (mL)	*p*-Value
Miyanji et al. [22]	2013	Pros. comp.	32	277	388	n/a ^a^
Miyanji et al. [23]	2015	Retro. comp.	46	261.5	471.1	0.000
Sarwahi et al. [24]	2016	Retro. comp.	22	600	800	0.051
Urbanski et al. [25]	2019	Retro. comp.	8	138.75 ^b^	450	0.016
Yang et al. [26]	2021	Retro. comp.	49	1279	2503	<0.001
Si et al. [27]	2021	Retro. comp.	112	502	808	<0.001
Sarwahi et al. [28]	2021	Retro. comp.	485	300	500	<0.001
Alhammoud et al. [29]	2022	Meta-analysis	107	271.1	527	0.019
Syundyukov et al. [40]	2023	Retro. comp.	82	208.7	564.3	<0.001
Sarwahi et al. [41]	2023	Retro. comp.	532	302 vs. 325 ^c^	500	0.005
Yang et al. [30]	2023	Meta-analysis	767	n/a ^d^		<0.001

The references number [25,27,40], specifically selected Lenke curves type 5C, 1-4, respectively 1. EBL = estimated blood loss; ml = milliliter; AIS = adolescent idiopathic scoliosis; SSI = segmental spinal instrumentation; PSF = posterior spinal fusion; MISS = minimally invasive spinal surgery; No. = number; OM = open posterior midline approach; pros. comp. = prospective comparative series; retro. comp. = retrospective comparative series; n/a = not available; CI = confidence interval; ^a^ = statistically significant difference: (95% CI −2.6 to −0.6); ^b^ = modified MISS technique using a single midline skin incision instead of three and fascial stab incisions for performing navigated SSI; ^c^ = 302 mL is related to the original MISS technique with three midline skin incisions; 325 mL is related to the modified MISS technique with a single midline skin incision; ^d^ = mean EBL expressed as WMD, −218.76; 95% CI −256.41 to 181.11.

## Data Availability

Not applicable.

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
