# Peer review of "Minimally Invasive Surgery for Posterior Spinal Instrumentation and Fusion in Adolescent Idiopathic Scoliosis: Current Status and Future Application"

_children, 2023, doi:10.3390/children10121882_

Round 1
Reviewer 1 Report
Comments and Suggestions for Authors
The Paper is a narrative review of MIS deformity surgery for AIS. The authors have comprehensively reviewed the published literature on the topic.
Comments:
1) Title :- Need to be move specific, to the topic description and needs to be edited - for regarding.
a) Minimally invasive posterior deformity correction in adolescent idiopathic scoliosis - A narrative review.
b) Minimally invasive surgery for posterior spinal instrumentation and fusion in Adolescent idiopathic scoliosis. - Current status and future application.
2) Abstract - Adequate
3) Over all the components of narrative review is well written, comprehensive and includes all the current published literature.
4) It would be beneficial to the readers if the authors could give same guidelines on the inclusion and exclusion crieteria for MIS deformity correction.
Author Response
Reviewer 1:
Comments and Suggestions for Authors
The Paper is a narrative review of MIS deformity surgery for AIS. The authors have comprehensively reviewed the published literature on the topic.
Comments:
1) Title :- Need to be move specific, to the topic description and needs to be edited - for regarding.
- a) Minimally invasive posterior deformity correction in adolescent idiopathic scoliosis - A narrative review.
- b) Minimally invasive surgery for posterior spinal instrumentation and fusion in Adolescent idiopathic scoliosis. - Current status and future application.
Ad comment Nr. 1: We agree with your comment and modified the title according to it. We chose the following new title: Minimally invasive surgery for posterior spinal instrumentation and fusion in Adolescent idiopathic scoliosis: Current status and future application.
2) Abstract - Adequate
Ad comment Nr. 2: Thank you for this positive comment.
3) Over all the components of narrative review is well written, comprehensive and includes all the current published literature.
Ad comment Nr. 3: Thank you for this positive comment.
4) It would be beneficial to the readers if the authors could give same guidelines on the inclusion and exclusion crieteria for MIS deformity correction
Ad comment Nr. 4: Thank you for this constructive comment. We consider that any Lenke type scoliosis can be treated using the posterior MISS approach. Also the papers we cite have very diverse recruitment selection criteria for their cohorts when it comes to Lenke classification. Some included only a specific type of Lenke curves in their study, like Syundyukov et al. (2023) and Yang et al. (2021) which selected Lenke type 1 curves or Urbanski et al. (2019), which selected Lenke type 5 curves, but other studies included any type of Lenke curves in their MISS groups. Therefore, and on the basis of our own experience, it appears that MISS procedures can be used for treating any type of Lenke curve. Concerning the curve magnitude and flexibility, we recommend to surgeons willing to adopt this technique to start with major curves under 70° and with a flexibility over 50%. However, with growing experience, the MISS approach might be used for greater and more fixed curves. We modified our revised manuscript as follows according to this comment: (p 10, lines 441 and 452-454).

Reviewer 2 Report
Comments and Suggestions for Authors
The aim of the manuscript “Minimally invasive surgery for posterior spinal instrumenta-tion and fusion in adolescent idiopathic scoliosis: results and benefits compared to the conventional approach” was to present the posterior MISS approach for performing PSF and SSI on AIS patients and compare its perioperative morbidity and radiological and clinical outcomes with those obtained using the traditional open posterior midline approach.
It is a descriptive review of a surgical technique for treating adolescent idiopathic scoliosis (AIS) using a minimally invasive spine surgery (MISS) approach, which is less invasive than the conventional open posterior midline approach.
The review provides a detailed description of the MISS technique, as well as a comprehensive comparison of its outcomes and complications with the open posterior midline approach, based on relevant publications and meta-analyses.
It concludes that MISS is a safe and effective alternative to the open posterior midline approach, and that it has advantages in terms of perioperative morbidity, blood loss, hospital stay, pain, and return to sports.
However, some weakness points can be identified as follows:
1. The review does not discuss the limitations or challenges of MISS, such as the technical difficulty, the learning curve.
2. It was not specified what type of scoliosis was discussed, e.g. according to Lenke?
3. In comparative studies on techniques, follow-up time is rarely provided, and when it is, it is usually not long-term.
4. The conclusions are quite subjective in favor of MISS. I recommend more objectivity.
Author Response
Reviewer 2:
Comments and Suggestions for Authors
The aim of the manuscript “Minimally invasive surgery for posterior spinal instrumenta-tion and fusion in adolescent idiopathic scoliosis: results and benefits compared to the conventional approach” was to present the posterior MISS approach for performing PSF and SSI on AIS patients and compare its perioperative morbidity and radiological and clinical outcomes with those obtained using the traditional open posterior midline approach.
It is a descriptive review of a surgical technique for treating adolescent idiopathic scoliosis (AIS) using a minimally invasive spine surgery (MISS) approach, which is less invasive than the conventional open posterior midline approach.
The review provides a detailed description of the MISS technique, as well as a comprehensive comparison of its outcomes and complications with the open posterior midline approach, based on relevant publications and meta-analyses.
It concludes that MISS is a safe and effective alternative to the open posterior midline approach, and that it has advantages in terms of perioperative morbidity, blood loss, hospital stay, pain, and return to sports.
However, some weakness points can be identified as follows:
- The review does not discuss the limitations or challenges of MISS, such as the technical difficulty, the learning curve.
Ad comment Nr.1:
Thank you for this comment. In our original manuscript, we mentioned the longer ORT associated with the MISS approach as a limitation (pages 8 and 9, lines 274-293) and also mentioned that this approach is technically more demanding (page 11, lines 384-385). The conclusion section of the original manuscript version however did not comprehensively discussed the challenges and limitations of this approach.
We therefore modified this section in the revised manuscript as follows (page 10, lines 450-457): However, the posterior MISS approach also has limitations. As the exposure is restricted in comparison to the traditional posterior midline approach, it is technically more challenging and associated with longer ORT. We therefore recommend to surgeons willing to adopt it, to exclude cases with a major curve over 70° or with less than 50% flexibility during their learning curve. According to Yang et al., which evaluated this learning curve effect in a recent case series including 76 AIS patients, a trained surgeon for conventional open scoliosis surgery needs to operate 46 cases using the MISS technique to reach proficient surgical skills.
- It was not specified what type of scoliosis was discussed, e.g. according to Lenke?
Ad comment Nr. 2:
Thank you for this comment. The majority of the cited studies do not select specific Lenke types of curves. Only references 23,25 and 33 selected some curve types according to Lenke. For more clarity, we now stated in the revised introduction section “The majority of the cited studies do not select specific Lenke types of curves. Should it be the case, it is explicitly stated where appropriate.” (page 2, lines 73-74). We further made changes according to this comment in the conclusion section (page 10, line 414) and where appropriate in the other sections of the revised manuscript (page 5, lines 199; page 5, lines 206-207; page 6, line 244; page 7, line 296; page 9, lines 377-378; page 10, line 414).
- In comparative studies on techniques, follow-up time is rarely provided, and when it is, it is usually not long-term.
Ad comment Nr. 3:
Thank you for this comment. According to it, we now added the FU length for multiple citations throughout the revised manuscript (page 5, lines 184-185 and 188; page 6, line 228, line 232 and lines 265-266; page 9, line 400 and line 407; page 10, line 410-411, line 414, line 416-417 and line 447) and also added some general information about the FU length of the cited articles in the revised Abstract (page 1, lines 26-27).
- The conclusions are quite subjective in favor of MISS. I recommend more objectivity.
Thank you for this comment. As stated in our response to your comment Nr. 1, we mentioned the longer ORT associated with the MISS approach as a limitation (pages 8 and 9, lines 274-293) and also mentioned that this approach is technically more demanding (page 11, lines 384-385) in our original manuscript. However, the conclusion section did not clearly summarize the challenges and limitations of MISS. We therefore modified our conclusion section according to this comment as follows (page 10, lines 450-457): However, the posterior MISS approach also has limitations. As the exposure is restricted in comparison to the traditional posterior midline approach, it is technically more challenging and associated with longer ORT. We therefore recommend to surgeons willing to adopt it, to exclude cases with a major curve over 70° or with less than 50% flexibility during their learning curve. According to Yang et al., which evaluated this learning curve effect in a recent case series including 76 AIS patients, a trained surgeon for conventional open scoliosis surgery needs to operate 46 cases using the MISS technique to reach proficient surgical skills.

Reviewer 3 Report
Comments and Suggestions for Authors
Thank your for the opportunity to review this manuscript. I have some comments that must be addressed by authors:
1. The abstract should clarify what is meant by "short-term" and "medium-term" in the context of the follow-up outcomes to avoid vagueness.
2. The introduction seems to overstate the role of Suk et al.’s 2001 publication in popularizing the use of pedicle screws. While influential, it's unlikely that this single publication was solely responsible for popularizing the technique. Also, please refere to newere references.
3. The comparison in the introduction between Sarwahi et al.’s approach and the traditional approach appears to be based on only two case reports, which is a very small sample size to draw significant conclusions from.
4. The mention of "studies published over the past decade corroborated the postulated advantages" is vague.
5. introduction does not clearly establish the current state of knowledge or the specific gap in the literature that this review aims to address
6. The phrase "as they detailed it" in row 67 is somewhat informal and could be revised for a more academic tone
7. Surgical Techniques section provides a detailed description of various surgical techniques but does not clearly delineate the advantages or disadvantages of each approach in comparison to others.
8. There's a mention of "four times more radiation" with the use of CT-based navigation without providing context or reference to its safety or comparative risk
9. "Indeed" in line 376 is somewhat informal and could be replaced with a more formal transition.
10. "MISS is technically more challenging and requires a longer operative time when the original three skin incisions technique is used," is awkward and could be restructured
Author Response
Reviewer 3:
Comments and Suggestions for Authors
Thank your for the opportunity to review this manuscript. I have some comments that must be addressed by authors:
- The abstract should clarify what is meant by "short-term" and "medium-term" in the context of the follow-up outcomes to avoid vagueness.
Ad comment Nr. 1:
Thank you for this valuable comment. We now removed the vague terminology "short-term" and "medium-term" initially used in the original abstract and modified the revised abstract according to this comment as follows (page 1, lines 26-27): It specifically examines perioperative morbidity and radiological and clinical outcomes at a minimal follow-up length of 2 years (range 2-9 years).
- The introduction seems to overstate the role of Suk et al.’s 2001 publication in popularizing the use of pedicle screws. While influential, it's unlikely that this single publication was solely responsible for popularizing the technique. Also, please refere to newere references.
Ad comment Nr. 2:
Thank you for this valuable comment. According to it, we added more context and references including citations of more recent publications (Ref 2-8, 10-11, 14-15) in the revised introduction section (page 1, lines 33, 35, 37 and 40; page 2, lines 51 and 64). We also reformulated this section to moderate the role of Suk et al.’s 2001 publication in popularizing the technique (page 1, lines 37-38; page 2, lines 46-51).
- The comparison in the introduction between Sarwahi et al.’s approach and the traditional approach appears to be based on only two case reports, which is a very small sample size to draw significant conclusions from.
Ad comment Nr. 3:
Thank you for this comment. The cited article by Sarwahi et al. which was released in 2011 was a surgical technique paper intended to describe this new technique and to show its feasibility. We agree that it cannot be used to draw conclusions. The original version of the introduction section did not clearly explain that the potential advantages of MISS postulated in this citation were not exclusively based on the 2 included case reports but also on emerging evidence that was published at that time and was supporting minimal invasive spine surgery for treating adult spine deformity. The revised introduction section was modified to address this comment (page 2, lines 57-64).
- The mention of "studies published over the past decade corroborated the postulated advantages" is vague.
Ad comment Nr. 4:
Thank you for this comment. The revised introduction section was thoroughly modified according to it and no more contains this sentence (page 2, lines 65-70).
- introduction does not clearly establish the current state of knowledge or the specific gap in the literature that this review aims to address
Ad comment Nr. 5:
Thank you for this comment. The introduction section of the revised manuscript was thorouhgly modified according to this comment and also according to the comments Nr 2, 3 and 4 (pages 1-2, lines 32-74).
- The phrase "as they detailed it" in row 67 is somewhat informal and could be revised for a more academic tone
Ad comment Nr. 6:
We agree with this comment and modified this sentence as follows in the revised manuscript (see page 2, lines 79-81): The original Wiltse approach involved two paramedian skin incisions with bilateral paramedian incisions of the thoracolumbar fascia and bilateral blunt dissections to separate the multifidus and longissimus muscles.
- Surgical Techniques section provides a detailed description of various surgical techniques but does not clearly delineate the advantages or disadvantages of each approach in comparison to others.
Ad comment Nr. 7:
Thank you for this comment. The original surgical technique section describes MISS as it was reported by Sarwahi et al. in 2011. It also presents the possibility to use Gelpi retractors or tubular retractors and it further describes two approach modifications.
Concerning the choice of retractors, we consider Gelpi and tubular retractors as equivalent retractors and included this statement in the revised surgical technique section (page 3, lines 99-101).
One modification by Urbansky et al. uses CT-navigation and fascial stab incisions to even more reduce soft tissue dissection. The potential advantages and disadvantages of the approach modification by Urbansky et al. are exposed in the revised surgical technique section (page 3, lines 128-132) and in the revised perioperative morbidity section (page 7, lines 297-300). Another modification by Sarwahi et al. (2023 paper), is consisting in replacing the three small midine skin incisions by one longer midline incision. The advantages and disadvantages of this technique are already reported in various locations of the original manuscript (page 3, lines 133-137; page 7, lines 303-314; page 9, lines 380-383; page 10, lines 418-427). Therefore, no additional changes were performed concerning this second approach modification in the revised manuscript.
- There's a mention of "four times more radiation" with the use of CT-based navigation without providing context or reference to its safety or comparative risk
Ad comment Nr. 8:
Thank you for this comment. We added relevant information about the comparative safety of both techniques in the revised manuscript according to it (page 3, lines 127-132).
- "Indeed" in line 376 is somewhat informal and could be replaced with a more formal transition.
Ad comment Nr. 9:
We agree with this comment and modified the corresponding sentence as follows in the revised manuscript (page 10, line 441-443): The MISS approach has notably been shown to result in equivalent coronal deformity correction, with some evidence supporting better restoration of thoracic kyphosis.
- "MISS is technically more challenging and requires a longer operative time when the original three skin incisions technique is used," is awkward and could be restructured
Ad comment Nr. 10:
We agree with this comment and have withdrawn this sentence. We still kept the information that the newer SLIM Technique with one long incision is reducing significantly the ORT in comparison to the original MISS technique with three small skin incisions, in our revised manuscript (see page 8, lines 349-354 and page 11, lines 462-465).

Round 2
Reviewer 2 Report
Comments and Suggestions for Authors
no more comments.
Author Response
Thank you very much.